# Trauma Coagulopathy and Its Outcomes

**DOI:** 10.3390/medicina56040205

**Published:** 2020-04-24

**Authors:** Gabriele Savioli, Iride Francesca Ceresa, Sarah Macedonio, Sebastiano Gerosa, Mirko Belliato, Giorgio Antonio Iotti, Sabino Luzzi, Mattia Del Maestro, Gianluca Mezzini, Alice Giotta Lucifero, Elvis Lafe, Anna Simoncelli, Federica Manzoni, Lorenzo Cobianchi, Mario Mosconi, Fabrizio Cuzzocrea, Francesco Benazzo, Giovanni Ricevuti, Maria Antonietta Bressan

**Affiliations:** 1Emergency Department, Fondazione IRCCS Policlinico San Matteo, 27100 Pavia, Italy; irideceresa@gmail.com (I.F.C.); sarah.macedonio01@universitadipavia.it (S.M.); sebastiano.gerosa01@universitadipavia.it (S.G.); mita.bressan@gmail.com (M.A.B.); 2PhD School in Experimental Medicine, Department of Clinical-Surgical, Diagnostic and Pediatric Sciences, University of Pavia, 27100 Pavia, Italy; m.delmaestro@gmail.com; 3Intensive Care Unit, Fondazione IRCCS Policlinico San Matteo, 27100 Pavia, Italy; m.belliato@smatteo.pv.it (M.B.); g.iotti@smatteo.pv.it (G.A.I.); 4Neurosurgery Unit, Department of Surgical Sciences, Fondazione IRCCS Policlinico San Matteo, 27100 Pavia, Italy; sabino.luzzi@unipv.it; 5Neurosurgery Unit, Department of Clinical-Surgical, Diagnostic and Pediatric Sciences, University of Pavia, 27100 Pavia, Italy; gianluca.mezzini@yahoo.it (G.M.); alicelucifero@gmail.com (A.G.L.); 6Neuro Radiodiagnostic Department, Fondazione IRCCS Policlinico San Matteo, 27100 Pavia, Italy; e.lafe@smatteo.pv.it (E.L.); a.simoncelli@smatteo.pv.it (A.S.); 7Clinical Epidemiology and Biometry Unit, Fondazione IRCCS Policlinico San Matteo, 27100 Pavia, Italy; f.manzoni@smatteo.pv.it; 8General Surgery Unit, Fondazione IRCCS Policlinico San Matteo, 27100 Pavia, Italy; l.cobianchi@smatteo.pv.it; 9Orthopedics Unit, Fondazione IRCCS Policlinico San Matteo, 27100 Pavia, Italy; m.mosconi@smatteo.pv.it (M.M.); f.cuzzocrea@smatteo.pv.it (F.C.); f.benazzo@smatteo.pv.it (F.B.); 10Department of Internal Medicine and Therapeutics, Cellular Pathophysiology and Clinical immunology Laboratory, University of Pavia, 27100 Pavia, Italy; giovanni.ricevuti@unipv.it

**Keywords:** trauma, coagulopathy, hemodynamics, trauma management

## Abstract

*Background and Objectives:* Trauma coagulopathy begins at the moment of trauma. This study investigated whether coagulopathy upon arrival in the emergency room (ER) is correlated with increased hemotransfusion requirement, more hemodynamic instability, more severe anatomical damage, a greater need for hospitalization, and hospitalization in the intensive care unit (ICU). We also analyzed whether trauma coagulopathy is correlated with unfavorable indices, such as acidemia, lactate increase, and base excess (BE) increase. *Material and Methods:* We conducted a prospective, monocentric, observational study of all patients (*n* = 503) referred to the Department of Emergency and Acceptance, IRCCS Fondazione Policlinico San Matteo, Pavia, for major trauma from 1 January 2018 to 30 January 2019. *Results:* Of the 503 patients, 204 had trauma coagulopathy (group 1), whereas 299 patients (group 2) did not. Group 1 had a higher hemotransfusion rate than group 2. In group 1, 15% of patients showed hemodynamic instability compared with only 8% of group 2. The shock index (SI) distribution was worse in group 1 than in group 2. Group 1 was more often hypotensive, tachycardic, and with low oxygen saturation, and had a more severe injury severity score than group 2. In addition, 47% of group 1 had three or more body districts involved compared with 23% of group 2. The hospitalization rate was higher in group 1 than in group 2 (76% vs. 58%). The length of hospitalization was >10 days for 45% of group 1 compared with 28% of group 2. The hospitalization rate in the ICU was higher in group 1 than in group 2 (22% vs. 14.8%). The average duration of ICU hospitalization was longer in group 1 than in group 2 (12.5 vs. 9.78 days). Mortality was higher in group 1 than in group 2 (3.92% vs. 0.98%). Group 1 more often had acidemia and high lactates than group 2. Group 1 also more often had BE <−6. *Conclusions:* Trauma coagulopathy patients, upon arrival in the ER, have greater hemotransfusion (*p* = 0.016) requirements and need hospitalization (*p* = 0.032) more frequently than patients without trauma coagulopathy. Trauma coagulopathy seems to be more present in patients with a higher injury severity score (ISS) (*p* = 0.000) and a greater number of anatomical districts involved (*p* = 0.000). Head trauma (*p* = 0.000) and abdominal trauma (*p* = 0.057) seem related to the development of trauma coagulopathy. Males seem more exposed than females in developing trauma coagulopathy (*p* = 0.018). Upon arrival in the ER, the presence of tachycardia or alteration of SI and its derivatives can allow early detection of patients with trauma coagulopathy.

## 1. Introduction

Every year, there are ~6 million deaths worldwide because of traumatic injuries [1,2]. Although the problem mainly affects low- and middle-income countries, it is also found in high-income countries. For example, in Europe, traumatic injuries are the third-largest cause of mortality in the general population and the leading cause of mortality in young patients. They are also one of the main causes of disability, resulting in high direct and indirect costs [3]. Therefore, correct and quick identification of the cause of bleeding, as well as neglected coagulopathies, is of great importance for the correct management of several surgical pathologies [4,5,6,7,8,9,10,11,12,13,14,15,16,17,18,19,20,21].

Major trauma (MT) refers to an event that results in a single injury or multiple injuries of such magnitude that it constitutes a quoad vitam or quoad valetitudinem danger to the patient. Conventionally, trauma is defined as severe when the patient’s injury severity score (ISS) is >15 [3]. ISS calculation is possible only after the patient has undergone diagnostic investigations mainly in the hospital. To overcome this limitation, and as it is essential that a potential MT be recognized as soon as possible, in the extra hospital phase, triage criteria for MT [3] are used (Table 1). About 30% of MT patients seem to develop trauma coagulopathy upon arrival in the emergency room (ER) [22,23,24,25,26,27,28,29]. Although it was once believed that trauma coagulopathy begins hours or even days after the traumatic event, it is now clear that it begins at the moment of trauma [3]. About 40% of trauma deaths are the result of bleeding, 10% of which seem avoidable [30,31].

The early acute coagulopathy associated with traumatic injury has recently been recognised as a multifactorial primary condition that results from a combination of bleeding-induced shock, tissue injury-related thrombin–thrombomodulin complex generation, and the activation of anticoagulant and fibrinolytic pathways [18,19,20,21,24,27,28,29,30,31,32]. The severity of the coagulation disorder is influenced by environmental and therapeutic factors that result in, or at least contribute to, acidaemia, hypothermia, dilution, hypoperfusion, and coagulation factor consumption [18,19,27,33,34,35]. Moreover, the coagulopathy is modified by trauma related factors such as brain injury and individual patient-related factors that include age, co-morbidities, and pre-hospital fluid administration [35,36,37]. A number of terms have been proposed to describe the specific trauma-associated coagulopathic physiology, including acute traumatic coagulopathy [19,38], early coagulopathy of trauma [29], acute coagulopathy of trauma-shock [27], trauma-induced coagulopathy [39], and trauma-associated coagulopathy [40].

This study identified the correlation of trauma coagulopathy upon the patient’s arrival in the ER with hemotransfusion rate, mortality, or unfavorable outcomes. We also looked at whether there are factors associated with the development of trauma coagulopathy, such as more severe trauma (ISS), trauma involving multiple body districts, the trauma of some body districts in particular (such as head trauma, abdominal trauma), or even a condition of acidemia already present upon arrival in the ER.

Finally, we went to see if some vital parameters or indices from their derivatives can help to identify a possible trauma coagulopathy immediately upon entry into the ER.

## 2. Materials and Methods

### 2.1. Study Design

The present study was approved by the Internal Review Board of Fondazione IRCCS Policlinico San Matteo, Pavia, Italy (Prot #20180059069, Proc #20180017957, Approval date: 5 July 2018).

We conducted a prospective, monocentric, observational study of all patients referred to the Department of Emergency and Acceptance, IRCCS Fondazione Policlinico San Matteo, Pavia, for MT in 13 consecutive months (1 January 2018–30 January 2019). The primary focus was the feedback of trauma coagulopathy to arrival in the ER (expected 20%–25%). We evaluated the presence of trauma coagulopathy using traditional laboratory tests. Trauma coagulopathy patients were then recruited on the basis of changes in the international normalized ratio (INR) partial thromboplastin time (PTT), prothrombin time (PT), activated partial thromboplastin time (aPTT), or platelet count.

The primary outcome was the correlation of trauma coagulopathy with the mortality rate. Secondary outcomes were the correlation of trauma coagulopathy with the hospitalization rate, intensive care unit (ICU) rate, hemotransfusion rate, massive blood transfusion rate, and need for surgery during hospitalization. We also analyzed the correlation of trauma coagulopathy with pH, base excess (BE), lactates, the shock index (SI), the ISS, the number of body districts involved, or the involvement of some body districts in particular (head trauma, abdominal one). In addition, we investigated whether these indices, alone or in combination with each other or with trauma coagulopathy, are correlated with, as a primary endpoint, mortality and with, as secondary endpoints, the need for hemotransfusion or massive hemotransfusion, the hospitalization rate, the hospitalization rate in the ICU, and the need for surgery during hospitalization.

### 2.2. Inclusion and Exclusion Criteria

All patients referred to the ER and registered in the hospital’s MT registry were enrolled (*n* = 503). Of the 503 patients, 368 were men (73%) and 135 were women (27%).

Patients with two or more of the following criteria were categorized into the trauma coagulopathy group (group 1): aPTT > 32 s, PT < 70 s, INR > 1.3, PTT < 150 unit/microliter, fibrinogen > 200 mg/dL, and D-dimer > 500 ng/mL. Group 1 comprised 204 patients (164 men (80%) and 40 women (20%)).

Group 2 (no trauma coagulopathy) comprised the remaining 299 patients (204 men (68%) and 95 women (32%)).

### 2.3. Study Population

Eligible patients were reported and then identified in the electronic database through diagnosis codes of discharge corresponding to “polytrauma” or as a code of acceptance corresponding to triage of “major trauma” The reason is that, in the emergency setting of this pathology, contextual data collection would have involved the risk of taking time away from care.

The personal and clinical data of each patient were extracted using the PIESSE digital platform and medical records, when drafted. Each patient was individually examined to assess whether he or she could be included in the study. The data collected and reported by the emergency physician, clinical reports prepared by medical specialists, nursing diaries, and results and reports of laboratory and radiological examinations were included in the electronic folder. Demographics, causes, and dynamics of trauma; waiting time; process time; length of stay (LOS) in the DEA; time required to perform various instrumental examinations and reporting topics; vital parameters; means of arrival; entry and exit codes; and hematochemical and hemogasanalytic examinations were also recorded. In addition, the hospitalization rates, need for surgery during hospitalization, intensive care, and death rate were assessed. All performance folders were viewed and evaluated, and all computed tomography (CT) scans were thoroughly reviewed. All collected data were recorded using Microsoft Excel and subsequently used for statistical analysis. In total, we included 503 MT patients in this study.

### 2.4. Statistical Analysis

Statistical analyses were performed using STATA statistical software version 14 (Stata Corporation, TX, USA). Continuous variables were described with mean and standard deviation; qualitative variables were expressed as counts and percentages. Comparisons between two groups were performed using Student’s *t*-test and the Mann–Whitney nonparametric test. Associations between qualitative variables were evaluated using Fisher’s exact test. The correlation between continuous variables of interest was tested by calculating Spearman’s rank correlation coefficient (ρ).1 All tests were two-sided. *p* < 0.05 was considered statistically significant.

## 3. Results

### 3.1. Study Population

The average age of the 503 patients was 43 years, and 49% had ISS >16; the average ISS was 17.88.

#### 3.1.1. Group 1

The average age of group 1 (*n* = 204) was 44 years, with the highest peak in the 55–65 year age group. There is a clear prevalence of males (79% vs. 21% in group 1; 68% vs. 32% in group 2), which is more pronounced in group 1, with statistical significance, compared with group 2 (*p* = 0.018). In group 1, 91% of the patients had trauma according to dynamic criteria, 22% according to anatomical criteria, and 15% according to clinical criteria. In addition, 46% required complete activation of the trauma team, whereas 12% required only partial activation (Figure 1). Vital parameters (blood pressure, oxygen saturation, Glasgow coma score (GCS) were found to be comparable in the two groups, except for the higher prevalence of tachycardia (>110 bpm) in group 1 (12.9% in group 1 vs. 5.1% in group 2; *p* = 0.008). Descriptively, the average heart rate (HR) was 87 bpm; 12.9% of the patients had a high HR (>110 bpm). The average systolic blood pressure (SBP) was 129 mmHg; 10% of the patients had SBP >90 mmHg. The average differential pressure was 54 mmHg; 1.27% of the patients had a differential pressure of <20 mmHg. The average saturation was 97.88. In addition, 6% of the patients had a Glasgow coma score (GCS) of 14–15, 3.6% had GCS between 9 and 13, and 7.8% had GCS <9.

In descending order, the causes of trauma were road accidents in 68% of the patients, domestic accidents in 8%, self-inflicted violence in 3%, work accidents in 3%, assault in 0.5%, and other causes in 5%. There is no statistically significant difference in the various causes of trauma between the two groups. In both, there is a clear prevalence of road accidents (Figure 2).

With regard to the way patients reached the ER, 9% were transported by helicopter rescue, 74% by an advanced rescue ambulance, and 70% by a basic rescue ambulance, whereas 2.5% arrived on their own (Figure 3).

With regard to the body districts affected, head trauma was the most frequent injury (59%), followed by chest trauma (48.5%), spine trauma (37.5%), lower limb trauma (27.5%), upper limb trauma (25%), pelvic trauma (18%), and abdominal trauma (17.5%). In 24% of the patients, only one body district was affected; 35%, two body districts; 20%, three body districts; 10%, four body districts; and 17%, more than four body districts. The number of districts involved is higher, and statistically significant, in group 1 (*p* = 0.000). With regard to the districts involved, only head trauma (*p* = 0.000) and abdominal trauma (*p* = 0.057) are related with the development of the major trauma clot.

#### 3.1.2. Group 2

The average age of group 2 was 43 years, with the highest peak in the 25–35 year age group. Of the 299 patients, 91% had trauma according to dynamic criteria, 10% according to anatomical criteria, and 8% according to clinical criteria. In addition, 45% required complete activation of the trauma team, whereas 6% required only partial activation (Figure 1). The average HR was 84 bpm; 5.17% of the patients had a high HR (>110 bpm). The average SBP was 132 mmHg; 5% of the patients had SBP <90 mmHg. The average differential pressure was 77 mmHg; no patient had a differential pressure of <20 mmHg. The average saturation was 98.95. In addition, 9% of the patients had GCS of 14–15, 1.6% had GCS between 9 and 13, and 2.5% had GCS <9.

In descending order, the causes of trauma were road accidents in 77% of the patients, domestic accidents in 8%, self-inflicted violence in 2%, work accidents in 6%, assault in 0.3%, and other causes in 3% (Figure 2).

With regard to the way patients reached the ER, 6% were transported by helicopter rescue, 64% by an advanced rescue ambulance, 39% by a basic rescue ambulance, and 4% by a nursing rescue ambulance, whereas 7% arrived on their own (Figure 3). There is no statistically significant difference between the two groups even for the way patients reached the ER.

With regard to the body districts affected, thoracic trauma was the most frequent (43%), followed by head trauma (37%), spine trauma (32%), lower limb trauma (25%), upper limb trauma (24%), abdominal trauma (14%), and pelvic trauma (12%). In 42% of the patients, only one body district was affected; 29%, two body districts; 14%, three body districts; 5%, four body districts; and 4%, more than four body districts.

### 3.2. Outcomes

#### 3.2.1. Hemotransfusion Rate

Group 1 had a higher hemotransfusion rate than group 2. This was true for both transfusions needed in the ER (8.5% vs. 3.3%; *p* =0.028) (Figure 4) and transfusions needed during hospitalization (11.32% vs. 4%; *p* = 0.016) (Figure 5)

#### 3.2.2. Hemodynamic Instability

In group 1, 15% of the patients showed hemodynamic instability compared with only 8% of the patients in group 2.

We considered SI <0.7 as indicative of the absence of shock (class I), SI between 0.7 and 1 as indicative of minor shock (class II), SI between 1 and 1.4 as indicative of moderate shock (class III), and SI >1.4 as indicative of severe shock (class IV). The SI distribution was worse in group 1 than in group 2. The layered SI was significantly higher in group 1 than in group 2 (*p* = 0.016). In addition, the modified SI was more frequently altered in group 1 than in group 2 (9.8% vs. 4.76%; *p* = 0.0051). Group 1 also presented a more frequently altered age SI (>50) than group 2 (7.84% vs. 5.49%; *p* = 0.075) (Table 2).

Group 1 was more often hypotensive (SBP <90 mmHg; *p* = 0.051), tachycardic (HR >110 bpm; *p* = 0.006), and with low oxygen saturation (sO2 <95% mmHg; *p* = 0.007) compared with group 2.

#### 3.2.3. Anatomical Outcomes

In group 1, 22% of the patients met the anatomical criteria for the definition of severe trauma compared with only 10% of the patients in group 2. Group 2 had a more severe ISS than group 2 (>16 in 21.26% vs. 15.7%; *p* = 0.0000) (Table 2).

The number of body districts affected was higher in group 1 than in group 2: 47% of group 1 had three or more body districts involved compared with 23% of group 2 (*p* = 0.000). In addition, 88.6% of group 1 had GCS of 14–15, 3.6% had GCS between 9 and 13, and 7.8% had GCS <9. In contrast, 95.9% of group 2 had GCS of 14–15, 1.6% had GCS between 9 and 13, and 2.5% had GCS <9.

#### 3.2.4. Use of Resources

The hospitalization rate was higher in group 1 than in group 2 (76% vs. 58%; *p* = 0.032). The length of hospitalization was >10 days for 45% of group 1 compared with 28% of group 2 (*p* = 0.245) (Table 3).

The hospitalization rate in the ICU was higher in group 1 than in group 2 (22% vs. 14.8%; nonsignificant *p* value). The average duration of ICU hospitalization was longer in group 1 than in group 2 (12.5 vs. 9.78 days; *p* = 0.100) (Table 3).

The need for surgery during hospitalization was comparable between both groups (19.8% vs. 17.2%) (Table 3).

#### 3.2.5. Mortality

Mortality was higher in group 1 than in group 2 (3.92% vs. 0.98%). The figure does not reach statistical significance (*p* = 0.168).

#### 3.2.6. Correlation with Other Unfavorable Outcomes

Arterial blood gas analysis showed that group 1 more often had acidemia (5.5% vs. 1.32% with pH < 7.35; *p* = 0.0142) and high lactates than group 2 (29.5% vs. 2.32% with lactates >1.9 mmol/L; *p* = 0.00001). Group 1 also more often had BE <−6 than group 2 (1.97% vs. 0%; *p* = 0.0262) (Table 4).

## 4. Discussion

Patients with trauma coagulopathy upon arrival in the ER have a significantly greater need for hemotransfusion than patients without trauma coagulopathy, which might be related to trauma coagulopathy itself. Several studies agree with this finding [1,12]. The reason is likely the hemorrhagic consequences of trauma coagulopathy and the state of hypocoagulability that characterizes trauma coagulopathy, which, as mentioned before, is one of the main causes of MT-related mortality [32].

It is also true, however, that trauma coagulopathy patients generally report higher severity of the trauma. We have verified that trauma coagulopathy patients have a significantly higher ISS than patients without trauma coagulopathy, as reported earlier [1,12]. The severity of the trauma is, therefore, correlated with trauma coagulopathy, which is supported not only by a higher ISS, but also by the fact that the number of body districts involved in the trauma is also higher in trauma coagulopathy patients. This would indicate that the more severe the trauma, the more easily the patient can develop coagulopathy. It is interesting to note that the number of districts involved also seems to play a role in the development of coagulopathy. Our study data also show how head trauma and abdominal trauma seem to play an independent role in developing coagulopathy.

Instead, the cause of major trauma does not appear to play a real role in causing trauma coagulopathy.

Unlike other pathologies [24,33,34,35,36,37,38,39,40,41,42,43,44,45,46,47], the molecular mechanism underlying the pathophysiology of acquired coagulopathy is still not well known, although studies have reported that a few bioumoral parameters (pH, lactates, and BE) are significantly more compromised in trauma coagulopathy patients [24,32,48]. Because ours was not a pathophysiological study, we do not know whether this correlation stems from the fact that MT results more easily in coagulopathy or whether trauma coagulopathy increases the severity of the trauma itself or whether there is a mutual influence. However, the increased percentage of patients with acidosis immediately upon arrival in the ER in group 1 suggests that acidosis plays a role in the development of trauma coagulopathy from the beginning. Similarly, the higher prevalence of BE underlines the importance of having an early arterial blood gas and to remember in its interpretation that the rise in BE correlates with the development trauma coagulopathy. It would certainly be interesting to further investigate, from a pathophysiological point of view, this correlation that we found and have shown to be significant, because understanding the underlying mechanism would allow better management of trauma coagulopathy and especially trauma in general [1,28,48,49,50,51,52,53,54,55,56,57,58,59,60,61,62,63,64]. In both cases, this is reflected in a greater seriousness that leads to the need for hospital treatment. In fact, trauma coagulopathy patients have a significantly higher rate of hospitalization than ever before. Analysis of the need for hospitalization in the ICU, although higher in trauma coagulopathy patients, does not reach statistical significance. This fact could be owing to good management of bleeding, high adherence to specific guidelines, and decreased number of patients requiring such care. In this regard, there is weak statistical evidence for a need for longer resuscitation; the decreased number of patients requiring such hospitalization in the ICU is also likely owing to the low number of people admitted to the ICU. With regard to the need for surgery during hospitalization, there are no statistically significant differences between patients with and without trauma coagulopathy. The reason is likely the fact that good clotting management could help the patient avoid surgery, given that compliance with drug guidelines is very high (>95%) [65].

There are also no statistically significant differences in the outcomes between the two populations, which could probably be owing to low general intrahospital mortality. In this regard, studies have indicated a correlation of trauma coagulopathy with increased mortality [66], although some studies, such as ours, have found no correlation [67]. A larger number of patients and an additional enlistment period are needed in order to confirm the findings.

## 5. Conclusions

The study highlights how early analysis of the presence of trauma coagulopathy within a few minutes of arrival in the ER gives important indications of the patient’s outcome. In fact, patients who have early trauma coagulopathy need hemotransfusions (*p* = 0.016) and hospitalization (*p* = 0.032) more frequently. Trauma coagulopathy seems to be more present in patients with a higher ISS (*p* = 0.000) and a greater number of districts involved (*p* = 0.000). With regard to the corporate districts considered in its own right, it is evident as a role in the development of trauma coagulopathy can perhaps be assumed for head trauma (*p* = 0.000) and for abdominal trauma, which, in our study, almost achieves statistical significance (*p* = 0.057). Males seem more exposed than femines in developing trauma coagulopathy (*p* = 0.018). Upon arrival in the ER, the presence of tachycardia (12.9% in group 1 vs 5.1% in group 2; *p* = 0.008) or alteration of SI (*p* = 0.016), Modified Shock Index (MSI) (*p* = 0.009) and ageSI (*p* = 0.009) can allow early detection of patients with trauma coagulopathy.

## Figures and Tables

**Figure 1 medicina-56-00205-f001:**
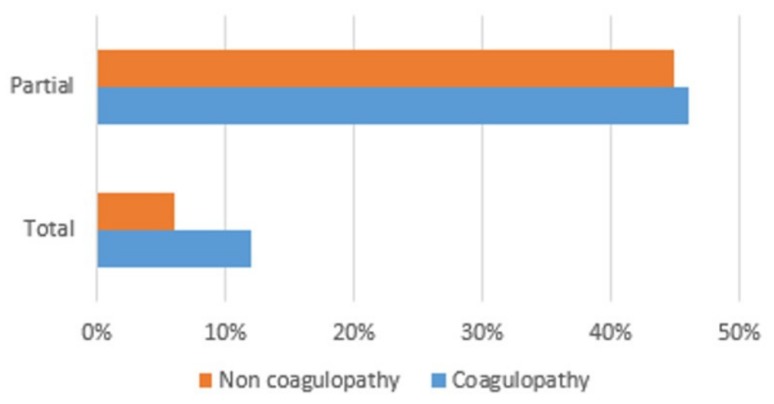
Trauma team activation.

**Figure 2 medicina-56-00205-f002:**
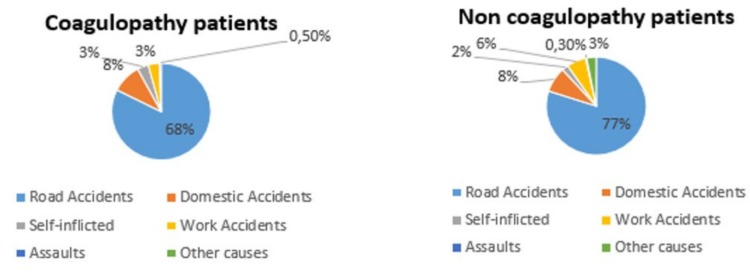
Causes of trauma.

**Figure 3 medicina-56-00205-f003:**
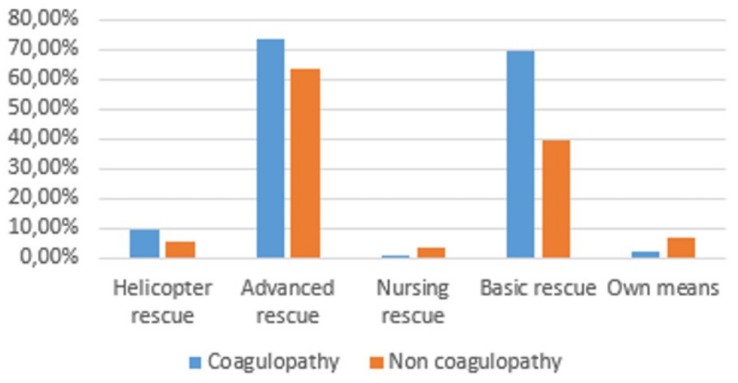
Arrival at the Emergency Room (ER).

**Figure 4 medicina-56-00205-f004:**
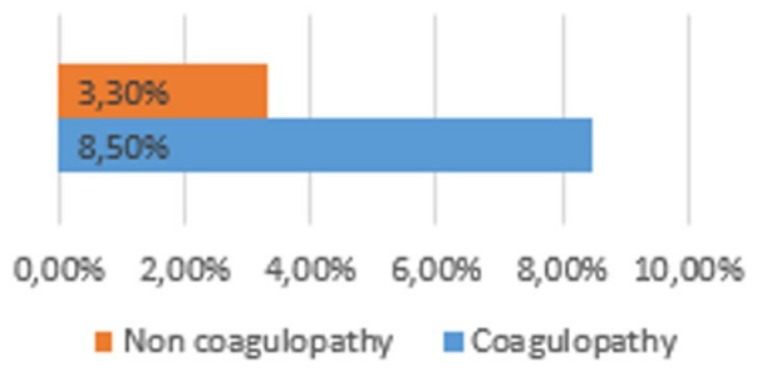
Haemotransusion rate.

**Figure 5 medicina-56-00205-f005:**
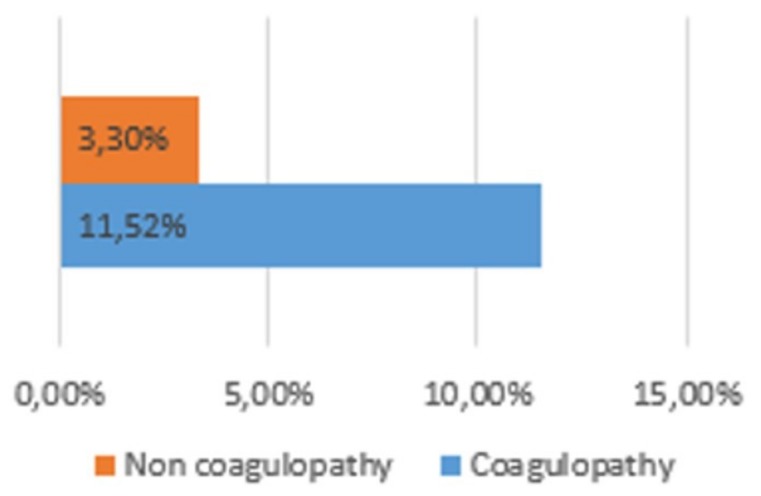
Intrahospital haemotransusion rate.

**Table 1 medicina-56-00205-t001:** Triage criteria for severe trauma. Criteria for activating the severe trauma protocol in our trauma center. Physiological, anatomical, dynamic criteria for defining probable severe trauma (one of the following criteria is sufficient).

Physiological Criteria	Anatomical Criteria	Dynamic Criteria
Eject from the vehicle	Penetrating head/neck/throat/abdomen/pelvic/armpit/groin trauma	Systolic blood pressure <90 mmHg
Motorcyclist thrown from the vehicle	Amputations above the wrist or ankle	Respiratory or breathlessness rate <10 or >29 acts/min
Died in the same vehicle	Chest trauma with flap/costal volet	State of consciousness (GCS) <13
Intruding of the cockpit >30 cm	Neurological injury with paralysis of even a single limb	
Fall from height >2 m	Fractures of two or more subxinextising bones	
Pedestrian projected or rolled or hit at speed >10 km/h	Suspected unstable fracture king of pelvis: Suspected unstable fracture	
High-energy impact (speed >65 km/h)	Skull fracture scuttled	
Vehicle coat	Burn >20% of body surface or airway/face	
Extrication time >20 min		

GCS, Glasgow coma score.

**Table 2 medicina-56-00205-t002:** Injury severity score (ISS) and shock index (SI).

	Coagulopathy	Non-Coagulopathy
ISS >15	21.26%	15.7%
MSI >1.3	9.8%	4.76%
ASI >50	7.84%	5.49%

MSI: Modified Shock Index ASI: Age Shock Index.

**Table 3 medicina-56-00205-t003:** Hospitalization. ICU, intensive care unit.

	Coagulopathy	Non-Coagulopathy	
% Hospitalization	76%	58%	*p* = 0.016
% ICU	22%	14.8%	
ICU Stay (day)	12.5%	9.8%	
% Operation theatre	19.2%	17.2%	
Intrahospital mortality	3.92%	0.98%	

**Table 4 medicina-56-00205-t004:** Biohumoral indices.

	Coagulopathy	Non-Coagulopathy
pH <7.35	5.5%	1.32%
% ICU	29.5%	2.32%
ICU Stay (day)	1.97%	0%

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
