# Peer review of "Trauma Coagulopathy and Its Outcomes"

_medicina, 2020, doi:10.3390/medicina56040205_

Round 1

Reviewer 1 Report

Savioli et al reported the significant impacts of coagulopathy on ICU stay, blood transfusion, and outcome in trauma. Those findings had been reported repeatedly and the authors need to show what they can add to the previous knowledge. The hot topic in this area is detecting the major factors that induce coagulopathy. As this is an observational study, it is not possible to discuss cause and result, but this reviewer suggests the authors approach this question with the possible measures. Is the brain injury an important factor, blood loss, unstable blood pressure, or acidosis a possible candidate? The authors mentioned the number of body district may relate to the presence of coagulopathy and that is an interesting finding.

Minor: the authors need to add units for the numbers (Page 2, bottom).

Author Response

Comments and Suggestions for Authors

Savioli et al reported the significant impacts of coagulopathy on ICU stay, blood transfusion, and outcome in trauma. Those findings had been reported repeatedly and the authors need to show what they can add to the previous knowledge. The hot topic in this area is detecting the major factors that induce coagulopathy. As this is an observational study, it is not possible to discuss cause and result, but this reviewer suggests the authors approach this question with the possible measures. Is the brain injury an important factor, blood loss, unstable blood pressure, or acidosis a possible candidate? The authors mentioned the number of body district may relate to the presence of coagulopathy and that is an interesting finding.

Response:

Kind reviewer. Thank you for encouraging us to make our messages clearer. His comments were extremely helpful and led to a wide-ranging review of the work, as you will see in the comments. We hope that we have correctly summarized them in the conclusions.

“The study highlights how early analysis of the presence of trauma coagulopathy within few minutes of arrival in E.R. gives important indications of the patient's outcome. In fact, patients who have early trauma coagulopathy need hemotransfusions (p=0.016) and hospitalization (p=0.032) more frequently. Trauma Coagulopathy seems to be more present in patients with a higher ISS (p=0.000) and a greater number of districts involved (p=0.000). With regard to the corporate districts considered in its own right, it is evident as a role in the development of Trauma Coagulopathy can perhaps be assumed for head trauma (p=0.000) and for abdominal trauma that in our study almost achieves the statistical significance (p.0.057). Males seem more exposed than femines in developing Coagulopathy trauma (p=0.018). Upon arrival in ER the presence of tachycardia (12.9% in group 1 vs 5.1% in group 2; p=0.008) or alteration of SI (p=0.016), MSI (p=0.009) and ageSI (p=0.009). can allow early detection of patients with Trauma Coagulopathy”

instead of the previous one “: Trauma coagulopathy patients, upon arrival in the ER, have greater hemotransfusion requirements than patients without trauma coagulopathy. They are also more hemodynamically unstable, present more severe anatomical damage, and need hospitalization”

More specifically, thanks to his suggestion, we thought we'd change:

  • In abstract the previous conclusion in “Trauma coagulopathy patients, upon arrival in the ER, have greater hemotransfusion (p=0.016) requirements and need hospitalization (p=0.032) more frequently than patients without trauma coagulopathy. Trauma Coagulopathy seems to be more present in patients with a higher ISS (p=0.000) and a greater number of anatomical districts involved (p=0.000). Head trauma (p=0.000) and abdominal trauma (p.0.057) are related to the development of Trauma Coagulopathy. Males seem more exposed than femines in developing Trauma Coagulopathy (p=0.018). Upon arrival in ER the presence of tachycardia or alteration of SI and its derivatives can allow early detection of patients with Trauma Coagulopathy.”
  • In line 131 after “The average age of group 1 (n = 204) was 44 years, with the highest peak in the 55–65 year age group.” we propose the addition: “there is a clear prevalence of males (79% vs 21% in group 1; 68% vs 32% in group 2), this is more pronounced than in group 1, with statistical significance compared to group 2 (p=0.018).
  • In line 133 after “ In group 1, 91% of the patients had trauma according to dynamic criteria, 22% according to anatomical criteria, and 15% according to clinical criteria. In addition, 46% required complete activation of the trauma team, whereas 12% required only partial activation.” we propose the addition: “vital paramers (blood pressure, oxygen saturation, CGS) were found comparable in the two groups, except for the higher prevalence of tachycardia (>110 bpm) in group 1 (12.9% in group 1 vs 5.1% in group 2; p=0.008).
  • to change “Trauma coagulopathy patients have” (line 227) in “Patients with Trauma Coagulopathy upon arrival in ER” to make it clear that our study analyzes the presence of Trauma Coagulopathy as early as arrival in E.R.
  • In line 237, after “coagulopathy patients”. we would like to add: " this would indicate how the more severe the trauma is, the more easily can develop Coagulopathy. It is interesting to note that the number of districts involved also seems to play a role in the development of Coagulopathy. Our study data also shows how head trauma and abdominal trauma seem to play an independent role in developing Coagulopathy. Instead, the cause of major trauma does not appear to play a real role in causing trauma coagulopathy”.
  • In line 149, after “more than four body districts” we propose the addition: “the number of districts involved is higher, statistically significant, in group 1 (p=0.000). (vd table....). with regard to the districts involved only the head trauma (p=0.000) and abdominal trauma (p.0.057) are related with the development of the major trauma clot.
  • In line 243 after “Since ours was not a pathophysiological study, we do not know whether this correlation stems from the fact that MT results more easily in coagulopathy or whether trauma coagulopathy increases the severity of the trauma itself or whether there is a mutual influence” we propose the addition: “However, the increased percentage of patients with acidosis immediately upon arrival in ER in group 1 suggests that acidosis plays a role in the development of trauma coagulopathy from the beginning. Similarly, the higher prevalence of BE underlines the importance of having an early EAB and to remember in its intepretation that the rise in BE correlates with correlates with the development trauma coagulopathy.

We hope in this way that we have emphasized, thanks to your valuable commentary, what data of our study highlights and what highlights compared to the previous knowledge. Particularly what factors appear to be related to coagulopathy (higher ISS score, more body districts involved, the presence of head trauma and the presence of abdominal trauma). Unfortunately, we did not have great results to report on unstable pressure, but instead we photographed how SI and its derivatives calculated immediately with the first vital parameters measured to the patient allow to highlight and recognize patients who will most easily develop Trauma Coagulopathy.

A characteristic point of our first study is to highlight how the very first patient assessment is already and poses to be of great help in stratification of the risk of worse outcomes.

Minor: the authors need to add units for the numbers (Page 2, bottom).

aPTT > 32 seconds, PT < 70 seconds, INR > 1.3, PTT < 150 unit/microliter, fibrinogen > 200 mg/dL, and D-dimer > 500 ng/mL

Reviewer 2 Report

The authors present a single-center observational study of trauma coagulopathy in trauma patients and different outcomes related to it.

  1. Under introduction, Line 49, 64-65 needs reference.
  2. Line 88- % of men and women don't add up to 100%.
  3. Line 90, I think PT > 120 is a typo?
  4. Line 194, another typo - Group 2 had a more severe Iss than group 2 ?
  5. Data presented regarding causes of trauma and arrival at the ED dont have any conclusion ? Please mention if there were any conclusions drawn from the data.
  6. There is not much difference in the two groups regarding hospitalization . So the conclusion that trauma coagulopathy patients have greater need for hospitalization in ICU is not true.
  7. With regards to trauma coagulopathy group itself, there is very little information present. A lot more information about that group, if available, can make the paper stronger, such as, outcomes of that group based on severity of the coagulopathy/standard tests that were performed.

Author Response

Dear Reviewer

We thank you very much for encouraging us to improve the parts of our work that were not sufficiently clear or not adequately developed. He has given us, with his suggestions, the possibility of a wide-ranging review that we hope fully responds to his suggestions. Conclusions have been developed and made more obvious and more appropriate.

In particular:

  • Under introduction, Line 49, 64-65 needs reference.

thank you for reporting it.

For the reference of Line 49:

-. World Health Organization. Injuries and violence: the facts; http://whqlibdoc.who.int/publications/2010/9789241599375_eng.pdf; 2010. Accessed 30 Jan 2015

-. GBD 2013 Mortality and Causes of Death Collaborators. Global, regional, and national age–sex specific all-cause and cause-specific mortality for causes of death, 1990–2013: a systematic analysis for the Global Burden of Disease Study 2013. Lancet. 2015;385(9963):117–71.

For the reference of Line 64-65:

-. Nicola Curry, Sally Hopewell, Carolyn Dorée, The acute management of trauma hemorrhage: a systematic review of randomized controlled trials. Crit Care. 2011; 15(2): R92.

-. Rolf Rossaint, Bertil Bouillon, Vladimir Cerny, The European guideline on management of major bleeding and coagulopathy following trauma: fourth edition Crit Care. 2016; 20: 100. Published online 2016 Apr 12. doi: 10.1186/s13054-016-1265-x PMCID: PMC4828865 PMID: 27072503

we also propose the addition for line 62 of the following references:

- Frith D, Goslings JC, Gaarder C, Maegele M, Cohen MJ, Allard S, et al. Definition and drivers of acute traumatic coagulopathy: clinical and experimental investigations. J Thromb Haemost. 2010;8(9):1919–25.

- Maegele M, Lefering R, Yucel N, Tjardes T, Rixen D, Paffrath T, et al. Early coagulopathy in multiple injury: an analysis from the German Trauma Registry on 8724 patients. Injury. 2007;38(3):298–304.

- Brohi K, Singh J, Heron M, Coats T. Acute traumatic coagulopathy. J Trauma. 2003;54(6):1127–30.

MacLeod JB, Lynn M, McKenney MG, Cohn SM, Murtha M. Early coagulopathy predicts mortality in trauma. J Trauma. 2003;55(1):39–44.

- -Schöchl H, Nienaber U, Maegele M, Hochleitner G, Primavesi F, Steitz B, et al. Transfusion in trauma: thromboelastometry-guided coagulation factor concentrate-based therapy versus standard fresh frozen plasma-based therapy. Crit Care. 2011;15(2):R83.

- Schöchl H, Frietsch T, Pavelka M, Jambor C. Hyperfibrinolysis after major trauma: differential diagnosis of lysis patterns and prognostic value of thrombelastometry. J Trauma. 2009;67(1):125–31.

Maegele M, Schochl H, Cohen MJ. An update on the coagulopathy of trauma. Shock. 2014;41 Suppl 1:21–5.

- Khan S, Davenport R, Raza I, Glasgow S, De’Ath HD, Johansson PI, et al. Damage control resuscitation using blood component therapy in standard doses has a limited effect on coagulopathy during trauma hemorrhage. Intensive Care Med. 2015;41(2):239–47.

In addition, thanks to his suggestion to improve the introduction, we propose to include at line 66: “The early acute coagulopathy associated with traumatic injury has recently been recognised as a multifactorial primary condition that results from a combination of bleeding-induced shock, tissue injury-related thrombin-thrombomodulin-complex generation and the activation of anticoagulant and fibrinolytic pathways [18–21, 24, 27–32]. The severity of the coagulation disorder is influenced by environmental and therapeutic factors that result in, or at least contribute to, acidaemia, hypothermia, dilution, hypoperfusion and coagulation factor consumption [18, 19, 27, 33-35]. Moreover, the coagulopathy is modified by traumarelated factors such as brain injury and individual patient-related factors that include age, co-morbidities, and pre-hospital fluid administration [35-37]. A number of terms have been proposed to describe the specific trauma-associated coagulopathic physiology, including Acute Traumatic Coagulopathy [19, 38], Early Coagulopathy of Trauma [29], Acute Coagulopathy of Trauma-Shock [27], Trauma-Induced Coagulopathy [39] and Trauma-Associated Coagulopathy [40].” (references reported in the draft)

  • Line 88- % of men and women don't add up to 100%.

Thank you for reporting it.

Women 27%. I'm sorry for typo

  • Line 90, I think PT > 120 is a typo?

      Thank you for reporting it.

      Women 27%. I'm sorry for typo

  • Line 194, another typo - Group 2 had a more severe Iss than group 2 ?

Thank you for reporting it.

Group 1 had a more severe ISS than group 2 (>16 in 21.26% vs. 15.7%; P =.0000).

I'm so sorry for typo.

  • Data presented regarding causes of trauma and arrival at the ED dont have any conclusion ? Please mention if there were any conclusions drawn from the data.

Thank you for the comment that allows us to emphasize this point.

There is no statistically significant difference in the various causes of trauma between the two groups. In both, there is a clear prevalence of road accidents. Unlike the severity (ISS scale) and the number of anatomical districts involved, the cause of major trauma does not appear to play a real role in causing trauma coagulopathy.

so we propose to add in line 141: “there is no statistically significant difference in the various causes of trauma between the two groups. In both, there is a clear prevalence of road accidents.” And in line 169 “there is no statistically significant difference between the two groups even for the way patients reached the E.R.”

  • There is not much difference in the two groups regarding hospitalization . So the conclusion that trauma coagulopathy patients have greater need for hospitalization in ICU is not true.

He's absolutely right. it was a mistake. The sentence must be corrected.

There is a significant difference regarding hospitalization in the two groups but not for hospitalization in ICU. I apologize for refusing.

  • With regards to trauma coagulopathy group itself, there is very little information present. A lot more information about that group, if available, can make the paper stronger, such as, outcomes of that group based on severity of the coagulopathy/standard tests that were performed.

kind reviewer, thank you very much for this comment. his suggestion will be a guide to an upcoming job, with a larger case. In fact, we think that in order to better stratify the coagulopathy group, we need larger numbers and specific work. We have made it clearer - as you suggested - in the introduction that there is no single definition of Coagulopathy and that pathogenesis is not yet fully clarified. for these reasons we believe that it would be too arbitrary with the number of this study effectively stratify degrees of severity of Trauma Coagulopathy. We are already evaluating another specific study for this purpose, that also sees the trend over time of the indices themselves.

We hope to have anyway more connoting in the extensive review conducted by the group that has developed coagulopathy. In particular, in this work we have highlighted some factors that are more closely related to the development of coagulopathy. These are having suffered more severe trauma, having more districts involved, having suffered a head trauma or abdominal one, having already developed acidemia upon entry into ER. We have also connoted what is, we repeat, immediately at the entrance, the clinical picture of the patient who most easily develops Coagulopathy. we have highlighted how the first signals can be heart rate and SI calculation.

We hope that we have correctly summarized them in the conclusions.

“Trauma Coagulopathy seems to be more present in patients with a higher ISS (p=0.000) and a greater number of districts involved (p=0.000). With regard to the corporate districts considered in its own right, it is evident as a role in the development of Trauma Coagulopathy can perhaps be assumed for head trauma (p=0.000) and for abdominal trauma that in our study almost achieves the statistical significance (p.0.057). Males seem more exposed than femines in developing Coagulopathy trauma (p=0.018). Upon arrival in ER the presence of tachycardia (12.9% in group 1 vs 5.1% in group 2; p=0.008) or alteration of SI (p=0.016), MSI (p=0.009) and ageSI (p=0.009). can allow early detection of patients with Trauma Coagulopathy”.

We look forward to hearing from you at your earliest convenience

Gabriele Savioli and all the other Authors.

Round 2

Reviewer 1 Report

I think the paper was significantly improved. I have no further comment.

Author Response

I think the paper was significantly improved. I have no further comment.

We thank you very much for helping us improve our work.

Reviewer 2 Report

Overall changes makes the paper acceptable.

Author Response

Overall changes makes the paper acceptable.

We thank you very much for helping us improve our work.